# Phylogenetic Analysis of G and P Genotypes of Bovine Group A Rotavirus Strains Isolated from Diarrheic Vietnam Cows in 2017 and 2018

**DOI:** 10.3390/ani13142314

**Published:** 2023-07-14

**Authors:** Jihye Shin, Gyu-Nam Park, SeEun Choe, Ra Mi Cha, Ki-Sun Kim, Byung-Hyun An, Song Yi Kim, Soo Hyun Moon, Bang-Hun Hyun, Dong-Jun An

**Affiliations:** 1Virus Disease Division, Animal and Plant Quarantine Agency, Gimcheon 39660, Republic of Korea; shinji227@korea.kr (J.S.); changep0418@korea.kr (G.-N.P.); ivvi59@korea.kr (S.C.); rami.cha01@korea.kr (R.M.C.); kisunkim@korea.kr (K.-S.K.); songkim@korea.kr (S.Y.K.); msh103101@korea.kr (S.H.M.); hyunbh@korea.kr (B.-H.H.); 2College of Veterinary Medicine, Seoul University, Seoul 08826, Republic of Korea; anbh5043@gmail.com

**Keywords:** BoRVA, phylogenetic tree, G-type, P-type, cow

## Abstract

**Simple Summary:**

Group A rotaviruses (RVAs) are thought to be the major cause of neonatal calf diarrhea worldwide. RVA-induced diarrhea in calves results in large economic losses to farmers in terms of treatment costs and reduced weight gain. Recent studies reveal that bovine RVA is caused predominantly by G6P[1] in China, G10P[11] in Japan, and G6P[5] in South Korea, Bangladesh, and Uruguay. In Vietnam, most RVA studies have been conducted in humans, with little information available for viruses in livestock. Therefore, for the first time, we performed a genetic analysis of RVAs prevalent in Vietnamese cows.

**Abstract:**

This study aimed to investigate the genetic diversity of G- and P-type bovine RVAs (BoRVAs) prevalent in Vietnam. Between 2017 and 2018, the prevalence of BoRVAs detected in diarrhea samples from 8 regions was as low as 1.9% (11/582). The prevalence of the G-type was 45.5% for G6 and 18.2% for G10; however, 36.3% remain unidentified. Interestingly, all BoRVAs were investigated as P[11], and there was no diversity within this P-type. Geographically, the G6 and G10 types were not identified in any specific area; rather, they occurred in both Northern and Southern Vietnam. G6P[11] and G10P[11], which are combined G- and P-types, were identified in 71.4% and 28.6% of BoRVA-positive samples, respectively. Phylogenetic tree analysis revealed that the G6-type detected in Vietnamese cows is similar to strains derived from China, Japan, and Korea, whereas the G10 type is closely related to the Chinese strain. In addition, the P11 strain detected in Vietnamese cows is similar to the Spanish and Chinese strains. The BoRVA-positive rate was higher in cows aged less than 2 months (3.2%, 3/94) than in those aged 2 months or more (1.6%, 8/488). In summary, we detected the presence of G6P11 and G10P11 BoVRAs on Vietnamese cow farms, and found that they were more predominant in young calves than in older cows.

## 1. Introduction

Group A rotavirus (RVA; family *Reoviridae)* is a non-enveloped, triple-layered virus particle comprising 11 double-stranded RNA segments [1,2]. Bovine group A rotaviruses (BoRVAs) are one of the major diarrhea-causing pathogens that infect neonatal calves worldwide [3,4]. These viruses cause indirect economic losses to the beef and dairy industries due to high morbidity and mortality rates among infected calves [1,3,5,6]. RVA strains are classified according to the antigenic characteristics and the sequence of the genes encoding the 2 structural proteins: G-type for VP7 and P-type for VP4 [1]. At least 42 G-types (glycoprotein) and 58 P-types (protease-sensitive protein) have been identified worldwide in humans and animals. The following combined G- and P-types have been reported recently in many countries and are prevalent in cattle: G6P[1] in China and Brazil; G6P[5] in Korea and Australia; G6P[11] in Bangladesh and Uruguay; and G10P[11] in Japan [7,8,9,10,11,12,13].

Various genotypes of RVA have been investigated in animals and humans in Vietnam; indeed, approximately 46.8% of pediatric hospitalizations for acute diarrhea between 2013 and 2018 were caused by RVAs [14], and analysis of the genetic diversity among positive cases revealed prevalence of the G3P[8] type (43.1%), followed by G8P[8] (19.7%), G2P[4] (12.9%), and G1P[8] (12.9%) [14]. A previous study of the RVA genotype in Vietnamese children under 5 years old who were hospitalized with acute diarrhea between 2007 and 2008 identified G3P[8] (46.9%), G1P[8] (45.0%), G2P[4] (2.8%), and G9P[8] (0.6%) [15]. Although RVA causes significant economic losses to the livestock industry worldwide [16], few studies have examined the prevalence of animal RVAs in Vietnam. A previous study identified the genotypes of RVAs circulating in pigs in the Mekong Delta region of Vietnam; that study identified various combinations of 6 G-types (G2, G3, G4, G5, G9, and G11) and 4 P-types (P[6], P[13], P[23], and P[34]) [16]. Although the study found no association between RVAs and clinical symptoms accompanied by specific diarrheal symptoms [16], the authors argued that the endemic asymptomatic circulation of RVAs could further complicate the nature of rotavirus disease in pigs with diarrhea [16]. Rotaviruses are an important cause of diarrhea in calves as well as pigs, but few studies have attempted to examine bovine RVAs in Vietnam.

Here, we investigated the genotypic diversity of RVAs isolated from diarrhea fecal samples obtained from Vietnamese cows; the aim was to obtain information about the prevalence of RVAs in livestock.

## 2. Materials and Methods

### 2.1. Fecal Sample Collection from Cows and Real-Time RT-PCR of BoRVAs

A total of 582 cow diarrhea samples were collected from 8 regions across Vietnam in 2017 and 2018 (sample collection was aided by the veterinary medical college of Hanoi Agriculture University): 222 were collected in 2017 and 360 were collected in 2018. Of these, 92 were collected in Dien Bien, 71 in Phu Thọ, 89 in Vinh Phuc, 61 in Ha Noi, 84 in Bac Ninh, 78 in Nghe An, 55 in Ha Tinh, and 52 in Ho Chi Minh. It could not be determined whether samples were collected from cows vaccinated against BoRVAs. The samples were assigned to 1 of 2 groups based on age: 94 calves aged under 2 months and 488 aged over 2 months. Each cow fecal sample was diluted 1:3 in phosphate-buffered saline and centrifuged (3000× *g* for 20 min) to obtain a supernatant containing BoRVAs. The supernatant was filtered sequentially through 2 syringe filters (pore size 0.45 μm and 0.22 μm; Minisart^®^ Syringe Filter, Sartorius, Germany) and stored at −80 °C until required. RNA was extracted from the final filtered supernatant sample using an RNeasy mini kit (Qiagen Inc., Germantown, MD, USA). To screen for BoRVAs, the VP6 gene was amplified using real-time PCR [17]; a Ct value ≤ 33 was considered positive.

### 2.2. Genotyping and Sequencing of G- and P-Type BoRVAs

BoRVA-positive samples detected by real-time RT-PCR screening were used for G and P genotyping using the HelixCript™ easy cDNA synthesis kit (Nonohelix Co., Daejeon, Republic of Korea). An 887 bp VP7 gene fragment corresponding to the G genotype and a 664 bp VP4 gene fragment corresponding to the P genotype were detected using the Accu-Power^®^ ProFi Taq PCR PreMix kit (Bioneer Inc., Daejeon, Republic of Korea), used as described previously [18,19]. The amplified PCR products (VP7 and VP4 gene fragments) were subcloned into pGEM-T Vector System II (Promega, Madison, WI, USA), and sequenced with an ABI Prism 3730xl DNA sequencer (Cosmo Genetech Co., Seoul, Republic of Korea) using SP6 and T7 primers. The VP7 and VP4 gene sequences of BoRVA reference strains were obtained from the GenBank database, and genotyping (G and P) of these strains was performed using the web-based ViPR-tool (https://www.bv-brc.org, (accessed on 9 May 2023)) [20] and the nucleotide BLAST tool.

### 2.3. VP7 and VP4 Phylogenetic Trees

Multiple nucleotide sequence alignments of the detected Vietnamese BoRVAs with the reference BoRVA sequences were performed using CLUSTAL X (ver. 2.1). To ensure accurate phylogenetic tree analysis, the “find best model” function in MEGA X was used to identify an optimal nucleotide substitution model. Phylogenetic tree analysis of VP7 and VP4 was performed using the maximum likelihood (ML) method in MEGA X (Molecular Evolutionary Genetics Analysis X) software [21].

## 3. Results

### 3.1. Detection of BoRVA

The prevalence of BoRVA in samples collected from diarrheic cows across 8 regions in 2017 and 2018 was 1.9% (11/582) (Table 1). The yearly BoRVA-positive rates were 1.8% (4/222) in 2017 and 1.9% (7/360) in 2018 (Table 1). The geographical distribution of the 11 BoRVA-positive samples was as follows: 1 in Dien Bien, 1 in Phu Thọ, 2 in Vinh Phuc, 1 in Ha Noi, 2 in Bac Ninh, 1 in Nghe An, 1 in Ha Tinh, and 2 in Ho Chi Minh (Table 1). The BoRVA-positive rate according to age was 3.2% (3/94) in cows aged under 2 months, and 1.6% (8/488) in those aged over 2 months. The 3 strains isolated from calves aged under 2 months were BF310 (Dien Bien), BF315 (Bac Ninh), and BF555 (Ha Tinh) (Figure 1).

### 3.2. Combinations of G and P Genotypes

Of the 11 Vietnamese BoRVA strains isolated, 5 were G6, 2 were G10, and 4 were unconfirmed. Among these, G6 was the predominant VP7 genotype, accounting for 71.4% (5/7) of strains, whereas G10 accounted for 28.6% (2/7) (Figure 1). All VP4 genotypes were P[11] (Figure 1). Over the 2 years, the most common combination of G and P genotypes was G6P[11] (71.4%; 5/7), followed by G10P[11] (28.6%; 2/7) (Figure 1).

### 3.3. Phylogenetic Analysis of the G6 and G10 Genotypes

Phylogenetic tree analysis of the detected Vietnamese VP7 genes showed that 7 Vietnamese BoRVAs belonged to the G6 (n = 5) and G10 (n = 2) genotypes (Figure 2 and Figure 3). The G6 genotype of BoRVAs was divided into groups 1 and 2 (Figure 2). Group 1 belonged to Asian and American countries, whereas group 2 belonged to Asian, American, and European countries. Vietnamese BoRVAs of the G6 genotype belonged to both group 1 (BF250, BF310, and BF315 strains) and group 2 (BF580 and BF584 strains) (Figure 2). Group 1 strains were closely related to Korean strains (15CN01, KJ9-1, KNU-GJ10), whereas group 2 strains were closely related to Japanese strains (OKY102 and OKY109) (Figure 2). The G10 genotype was also divided into group 1 (which included most strains) and a minor group 2 (the E29TR strain from Turkey and the B8 strain from India) (Figure 3). The G10 genotype of Vietnamese BoRVAs belonged to group 1 (BF207 and BF275 strains) and was closely related to Chinese strains (SCMY1, SCMY3, SCMY6, SCMY7, SCMY9, SCMY10, SCMY11, SCMY20, XHX-07, HM26, and DQ75) (Figure 3). In addition, the following strains showed the highest homology with G6-type Vietnamese BoRVAs: the BF250 strain showed 99.8% homology with the 15CN01 strain (South Korea); the BF580 and BF 584 strains showed 94.5–95.8% homology with the FMV1089635 strain (Canada); and the BF310 and BF315 strains showed 98.3–98.6% homology with the KJ19-2 strain (South Korea). Vietnamese G10-type BoRVAs (BF207 and BF275) showed the highest homology (96.4–96.9%) with 2 Chinese strains (SCMY6 and SCMY7).

### 3.4. ML Tree Analysis of the P Genotype

The phylogenetic tree for VP4 revealed that the 11 Vietnamese BoRVAs (BF555, BF585, BF580, BF535, BF584, BF310, BF207, BF315, BF214, BF250, and BF272) were all of the same genotype, i.e., P[11] (Figure 4). The phylogenetic tree revealed that genotype P[11] was also divided into group 1 and group 2 (Figure 4). Group 2 included Korean strains (17GB11 and 18GB03) and Argentinian strains (B1191_B_BA, B1541, B1988_BA, and B996_B_Co) (Figure 4). Among Vietnamese rotaviruses belonging to Group 1, 6 strains (BF555, BF585, BF580, BF535, BF584, and BF310) were closely related to Spanish strains (116043 and 106775), while 5 strains (BF207, BF315, BF214, BF250, and BF272) were closely related to Chinese strains (SCMY2, SCMY9, HY-1, and HY-16) (Figure 4). Of the 11 P11-type Vietnamese BoRVAs, 6 (BF310, BF535, BF555, BF580, BF583, and BF584) showed 88.9–96.8% homology with strain R1WTA11 (Northern Ireland), whereas the remaining 5 (BF207, BF214, BF250, BF275, and BF315) showed 94.7–97.9% homology with Chinese strains (HY-1 and HY-16).

## 4. Discussion

A recent analysis of BoRVAs revealed that a combined genotype of the VP7 and VP4 genes emerged and became predominant in several Asian countries. For example, BoRVAs circulated widely among dairy calves in 2018, with G6P[1] being the dominant genotype in China [7]. A recent study from South Korea suggests that the G6P[5] genotype circulating among calves was most prevalent between 2014 and 2022 [8,22]. In Japan, BoRVAs circulating from 2018 to 2020 were predominantly G10P[11] (41.8%), with other genotypes (G6P[11], G6P[5], G6P[1], G8P[11], and G8P[1]) being detected at low rates [9]. In Bangladesh, G6P[11] was the predominant genotype of BoRVAs isolated from diarrhea samples between 2014 and 2015 (94.4%), followed by G10P[11] (5.6%) [10]. In Uruguay, the G-type and P-type combinations isolated from BoRVA-infected calves from dairy and beef herds between 2015 and 2018 were predominantly G6P[11] (40.4%) and G6P[5] (38.6%), with G10P[11] and G24P[33] being detected at much lower rates [11]. The G6-IV and P[5]-IX genotypes isolated predominantly from Brazilian calves in 2009 were investigated as major causes of diarrhea among BoRVA-vaccinated beef cow herds [12]. The most prevalent genotype of the BoRVA strains circulating in Australian calves during 2004 and 2005 was G6P[5] (38.5%) [13]. In several of the countries mentioned above, the G6P[1], G6P[5], G6P[11], and G10P[11] genotypes were the main G- and P-genotype combinations. In the present study, the main genotype isolated from infected Vietnamese cows was G6P[11], with G10P[11] being less common.

In Japan, China, and Korea, there have been changes in the prevalence of dominant G- and P-genotype combinations [7,9,22,23,24,25]. The predominant genotype of Japanese BoRVAs changed from G6P[5] to G10P[11] [9,26], that of Chinese BoRVAs changed from G6P[5] to G6P[1] [7,24,25], and that of Korean BoRVAs changed from G8P[7] to G6P[5] [22,23]. Unfortunately, because the present study is the first to investigate the genetics of BoRVA VP7 and VP4 genes in Vietnam, our data cannot be compared with those from past studies of Vietnamese BoRVAs. However, the rotaviruses (G6P[11] and G10[11]) identified in Vietnamese cows in this study should aid future research examining rotaviruses present in other livestock species in Vietnam.

The phylogenetic tree of VP7 genes revealed that the G10 genotype of Vietnamese BoRVAs belonged to group 1, while the G6 genotype belonged to both groups 1 and 2. This suggests genetic diversity; indeed, the Vietnamese BoRVA P[11] genotypes were also distant within group 1 of the phylogenetic tree for the VP4 gene. Different strains of Vietnamese BoRVAs show genetic diversity, even those within the G6 and P[11] genotypes. In particular, in the case of the G6 type, group 1 strains (BF250, BF310, and BF315) on the phylogenetic tree were detected in Northern Vietnam, and group 2 strains (BF580 and BF584) were detected in Central and Southern Vietnam. Analytical studies of previous VP7 genetic data confirm that the G6 genotype is divided into 5 lineages (I–V) [27,28,29,30]. A previous study of RVA strains in cows from Argentina showed that the Argentine G6 genotype belonged to lineages III and IV. The authors suggested that lineage IV is a typical bovine RVA strain, while lineage III also includes a human RVA strain [31]. Division of the G6 genotype into 5 lineages meant that the G6 genotype (BF250, BF310, BF315) detected in Northern Vietnam was included in lineage IV. In addition, the G6 genotype (BF580, BF584) detected in Southern and Central Vietnam was predicted to be included in lineage III or V.

Many studies worldwide report that the prevalence of BoRVAs in diarrheic calves ranges from 30–94% [32,33,34,35,36]. In particular, BoRVA infection is estimated to affect 27–36% of newborn calves [32], and, in some cases, BoRVA infection of newborn calves has a mortality rate of up to 80%; however, the mortality rate is usually around 5–20% [37]. Other studies from South Korea suggest that BoRVAs are an important cause of diarrhea in young calves (aged < 10 days) [8,22]. We found that the incidence of BoRVA infection in Vietnamese calves aged less than 2 months was twice that in adult cows; however, the prevalence of BoRVAs across 8 Vietnamese regions was 1.9%, which is very low when compared with data from other countries. Diarrhea in Vietnamese cows and calves is thought to be caused mainly by bacterial and parasitic infections rather than by BoRVAs. A previous study suggests that the rate of BoRVA infection in calves in temperate regions (e.g., China, Iran, Sweden, United Kingdom, France, Australia, Spain, Italy, Uruguay, Canada, and New Zealand) ranges from 13.9–79.9%, whereas that in tropical regions (e.g., India, Brazil, and Bangladesh) ranges from 11.8–25% [38]. These climatic considerations suggest that a temperate climate, well-defined seasons, and cold, dry winters are more suitable for BoRVA survival than perpetually hot tropical climates such as Vietnam [38]. In the future, continued monitoring is needed to clarify the cause of the low incidence of BoRVAs in Vietnamese calves.

## 5. Conclusions

Of the Vietnamese BoRVA strains detected, G6P[11] is the most prevalent, followed by G10P[11]. In addition, analysis of the VP7 gene sequences revealed that Vietnamese BoRVAs were similar to strains isolated from China, Korea, and Japan, whereas the VP4 gene showed high similarity with Spanish and Chinese strains. This study presents the first in-depth BoRVA genetic analysis of BoRVAs from Vietnamese cattle; thus the data will be used for future studies monitoring BoRVAs in Vietnam.

## Figures and Tables

**Figure 1 animals-13-02314-f001:**
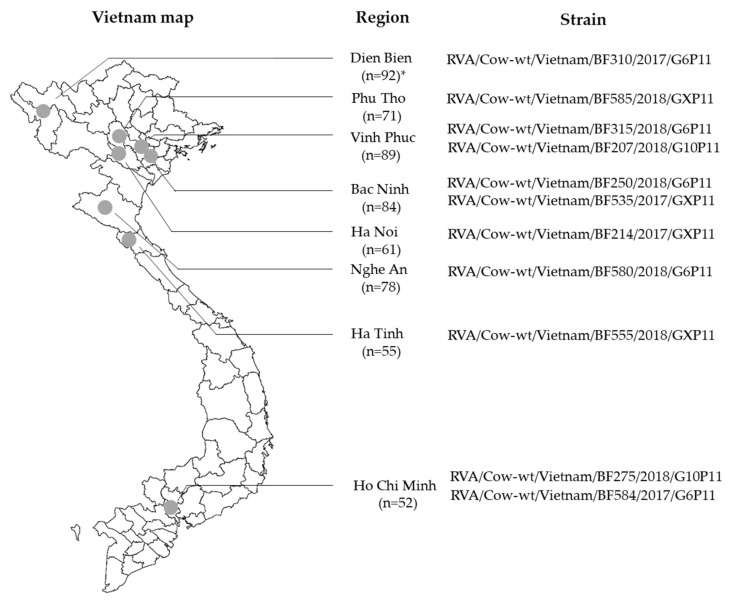
Areas in which bovine rotavirus was detected, along with the strain names. * Number of fecal samples collected from cows.

**Figure 2 animals-13-02314-f002:**
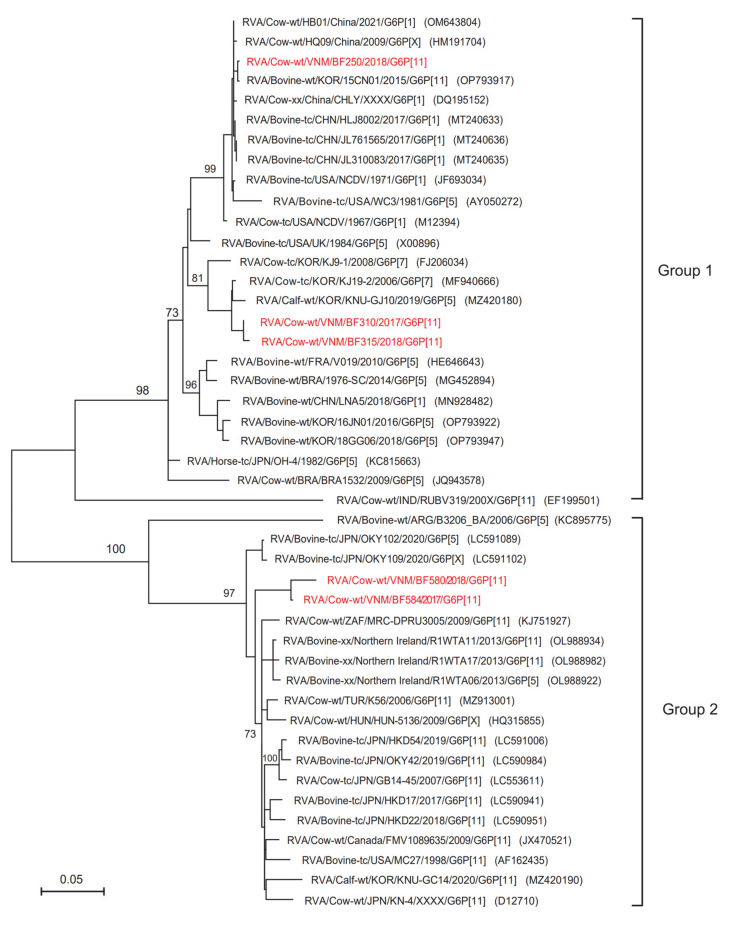
Phylogenetic analysis of the G6 genotype, based on the partial nucleotide sequence of the BoRVA VP7 gene. The phylogenetic tree was constructed for 45 BoRVAs (40 reference sequences and 5 Vietnam sequences) using the maximum likelihood method in the MEGA X program and the Tamura 3-parameter model, with bootstrap values of 1000. The five Vietnamese BoRVAs are highlighted in red.

**Figure 3 animals-13-02314-f003:**
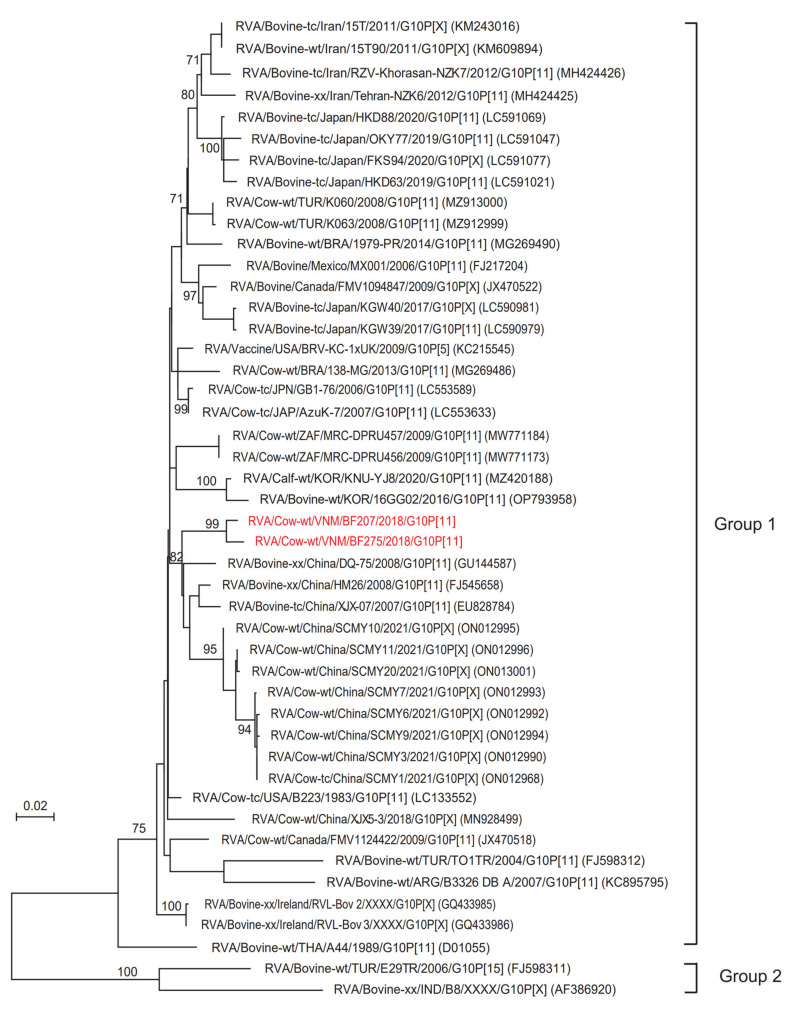
Phylogenetic analysis of the G10 genotype, based on the partial nucleotide sequences of the BoRVA VP7 gene. The phylogenetic tree was constructed for 46 BoRVAs (44 reference sequences and 2 Vietnam sequences) using the maximum likelihood method in the MEGA X program and the Tamura 3-parameter model, with bootstrap values of 1000. The two Vietnamese BoRVAs are highlighted in red.

**Figure 4 animals-13-02314-f004:**
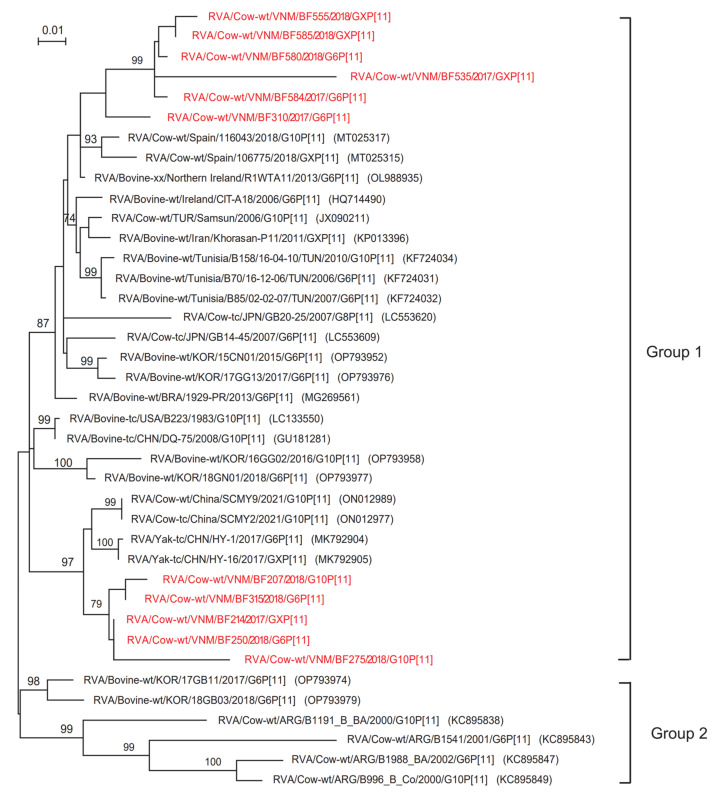
Phylogenetic analysis of the P[11] gene, based on the partial nucleotide sequence of the BoRVA VP4 gene. The phylogenetic tree was constructed for 39 BoRVAs (28 references and 11 Vietnam) using the maximum likelihood method in the MEGA X program and the Tamura 3-parameter model, with bootstrap values of 1000. The 11 Vietnamese BoRVAs are highlighted in red.

**Table 1 animals-13-02314-t001:** Prevalence of bovine rotavirus from cows in Vietnam in 2017 and 2018.

Year	North					Middle		South
	Dien Bien	Phu Thọ	Vinh Phuc	Ha Noi	Bac Ninh	Nghe An	Ha Tinh	Ho Chi Minh
2017	1 ^a^/44 ^b^	0/32	0/31	1/27	0/33	1/29	0/12	1/14
2018	0/48	1/39	2/58	0/34	2/51	0/49	1/43	1/38
Total	1.0% (1/92)	1.4% (1/71)	2.2% (2/89)	1.6% (1/61)	2.3% (2/84)	1.2% (1/78)	1.8% (1/55)	1.9% (2/52)

^a^ Number of BoRVA-positive samples; ^b^ number of cow fecal samples.

## Data Availability

The nucleotide sequences of the 2017–2018 viruses obtained in this study were submitted to the GenBank database under accession numbers OQ735383–OQ735384, OQ784921-OQ784925 for the VP7 gene, and OQ735385–OQ735395 for the VP4 gene.

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
