# Peer review of "Phylogenetic Analysis of G and P Genotypes of Bovine Group A Rotavirus Strains Isolated from Diarrheic Vietnam Cows in 2017 and 2018"

_animals, 2023, doi:10.3390/ani13142314_

Round 1

Reviewer 1 Report

In this paper, Shin et al., performed the first analysis of bovine rotaviruses (BoRVAs) in Vietnam. The authors detected 11 BoRVAs from 582 fecal samples by RT-PCR (VP6 gene). They used 5 of the 11 strains for further analyses. Here, 5 G6P[11] and 2 G10P[11] strains were identified. Phylogenetically, these Vietnamese BoRVA strains had genetic diversity on G6-VP7 and P[11]-VP4 genes. Although the information of G/P genotypes of Vietnamese BoRVAs is valuable, most importantly, more BoRVA strains and analyses (nucleotide sequence identity comparison and full genome-based analysis) are essential to draw the conclusion of this study, Unfortunately, the quality of the paper is still rather low. The language and sentence structure throughout this manuscript is slightly difficult to follow. This paper must be checked by a native English speaker and a senior virologist.   1.      Lines 22-23: …bovine RVAs (BoRVAs) prevalent… 2.      Lines 29-31: Every rotavirologist knows that BoRVAs have different G/P genotypes from those from HuRVAs and PoRVAs. 3.      Lines 32-33: This sentence is overstatement because authors analyzed just 7 BoRVA strains in this study. 4.      Line 43: “structural” instead of “VP” 5.      Line 44: Please update the numbers of G and P types 6.      Line 97: Please reconfirm the tool you used for RVA genotyping. 7.      Lines 108, 109, and 112: “Table 1” instead of “Table” 8.      Lines 121 and 123: “5/7” and “2/7” instead of “5/11” and “2/11” 9.      Lines 122-123: Please reconsider the sentence. 10.  Lines 132: “Asian” instead of “Asia” 11.  Figure 2: “RVA/Cow-wt/VNM/BF250/2018/G6P[11]” instead of “RVA/Cow-wt/Vietnam/BF250/2018/G6P[11]” ….. 12.  Line 157: …Vietnamese BoRVAs are … 13.  Line 179: What does the “BoRVA outbreaks in Asian countries” mean? 14.  Lines 211-213: Please reconsider the sentence. 15.  Lines 233-235: Authors concluded that RVAs endemic in Vietnamese cows are not related to RVAs in humans and pigs. However, the remaining 9 genes were not analyzed in this study. Full genome-based analysis is essential to conclude like that. 16.  References: At least, refs. 1 and 5 should be updated.

The language throughout this manuscript is slightly difficult to follow.

Author Response

In this paper, Shin et al., performed the first analysis of bovine rotaviruses (BoRVAs) in Vietnam. The authors detected 11 BoRVAs from 582 fecal samples by RT-PCR (VP6 gene). They used 5 of the 11 strains for further analyses. Here, 5 G6P[11] and 2 G10P[11] strains were identified. Phylogenetically, these Vietnamese BoRVA strains had genetic diversity on G6-VP7 and P[11]-VP4 genes. Although the information of G/P genotypes of Vietnamese BoRVAs is valuable, most importantly, more BoRVA strains and analyses (nucleotide sequence identity comparison and full genome-based analysis) are essential to draw the conclusion of this study, Unfortunately, the quality of the paper is still rather low. The language and sentence structure throughout this manuscript is slightly difficult to follow.

General comment: This paper must be checked by a native English speaker and a senior virologist.   

Answer: The manuscript has been revised by an English language editing company (www.bioedit.com).

Comment 1. Lines 22-23: …bovine RVAs (BoRVAs) prevalent…

Answer: Thank you. We have changed “bovine RVAs (BoRVA) prevalent in Vietnam” to “bovine RVAs (BoRVAs) prevalent in Vietnam” (Lines 22–23)

Comment 2. Lines 29-31: Every rotavirologist knows that BoRVAs have different G/P genotypes from those from HuRVAs and PoRVAs. 

Answer: We have revised the manuscript text as follows:

“Phylogenetic tree analysis revealed that the G6-type detected in Vietnamese cows is similar to strains derived from China, Japan, and Korea, whereas the G10 type is closely related to the Chinese strain. In addition, the P11 strain detected in Vietnamese cows is similar to the Spanish and Chinese strains. The BoRVA-positive rate was higher in cows aged less than 2 months (3.2%, 3/94) than in those aged 2 months or more (1.6%, 8/488).” (Lines 29–33).

Comment 3.  Lines 32-33: This sentence is overstatement because authors analyzed just 7 BoRVA strains in this study. 

Answer: The text has been revised as follows:

“In summary, we detected the presence of G6P11 and G10P11 BoVRAs on Vietnamese cow farms, and found that they were more predominant in young calves than in older cows.” (Lines 33–35).

Comment 4.  Line 43: “structural” instead of “VP” 

Answer: Thank you. We have changed “two VP proteins” to “two structural proteins” (Line 45).

Comment 5.  Line 44: Please update the numbers of G and P types.

Answer: We have revised the text as follows:

“At least 42 G-types (glycoprotein) and 58 P-types (protease-sensitive protein)” (Lines 45-46).

Comment 6.  Line 97: Please reconfirm the tool you used for RVA genotyping. 

Answer: The tool used was “ViPR-tool (https://www.bv-brc.org)” (Line 98).

Comment 7.  Lines 108, 109, and 112: “Table 1” instead of “Table” 

Answer: We have changed “Table” to “Table 1” (Lines 109, 110, 113).

Comment 8. Lines 121 and 123: “5/7” and “2/7” instead of “5/11” and “2/11” 

Answer: Thank you. We have revised the sentences as follows:

“71.4% (5/7) of strains, whereas G10 accounted for 28.6% (2/7)”

and

“G6P[11] (71.4%; 5/7), followed by G10P[11] (28.6%; 2/7)” (Lines 122–124).

Comment 9.  Lines 122-123: Please reconsider the sentence. 

Answer: We have revised the sentence as follows:

“Over the 2 years, the most common combination of G and P genotypes was G6P[11] (71.4%; 5/7), followed by G10P[11] (28.6%; 2/7) (Figure 1)” (Lines 123–124).

Comment 10. Lines 132: “Asian” instead of “Asia”

Answer: “Asia” has been changed to “Asian” (Line 133).

Comment 11.  Figure 2: “RVA/Cow-wt/VNM/BF250/2018/G6P[11]” instead of “RVA/Cow-wt/Vietnam/BF250/2018/G6P[11]” ….. 

Answer: We have changed “RVA/Cow-wt/Vietnam/BF250/2018/G6P[11]” to “RVA/Cow-wt/VNM/BF250/2018/G6P[11]”. In addition, all Vietnam strains in Figures 2, 3, and 4 have been changed to “VNM”.

Comment 12.  Line 157: …Vietnamese BoRVAs are … 

Answer: “Vietnamese BoRVA are” has been changed to “Vietnamese BoRVAs are” (Line 168).

Comment 13.  Line 179: What does the “BoRVA outbreaks in Asian countries” mean? 

Answer: This sentence has been changed to “A recent analysis of BoRVAs revealed that a combined genotype of the VP7 and VP4 genes emerged and became predominant in several Asian countries.” (Lines 185–186).

Comment 14.  Lines 211-213: Please reconsider the sentence. 

Answer: The manuscript text was been revised accordingly (lines 220–246).

Analytical studies of previous VP7 genetic data confirm that the G6 genotype is divided into five lineages (I–V) [27-30]. A previous study of RVA strains in cows from Argentina showed that the Argentine G6 genotype belonged to lineages III and IV. The authors suggested that lineage IV is a typical bovine RVA strain, while lineage III also includes a human RVA strain [31]. Division of the G6 genotype into five lineages meant that the G6 genotype (BF250, BF310, BF315) detected in northern Vietnam was included in lineage IV. In addition, the G6 genotype (BF580, BF584) detected in southern and central Vietnam was predicted to be included in lineage III or V. Many studies worldwide report that the prevalence of BoRVAs in diarrheic calves ranges from 30–94% [32-36]. In particular, BoRVA infection is estimated to affect 27–36% of newborn calves [32], and in some cases, BoRVA infection of newborn calves has a mortality rate of up to 80%; however, the mortality rate is usually around 5–20% [37]. Other studies from South Korea suggest that BoRVAs are an important cause of diarrhea in young calves (aged < 10 days) [8,22]. We found that the incidence of BoRVA infection in Vietnamese calves aged less than 2 months was twice that in adult cows; however, the prevalence of BoRVAs across eight Vietnamese regions was 1.9%, which is very low when compared with data from other countries. Diarrhea in Vietnamese cows and calves is thought to be caused mainly by bacterial and parasitic infections rather than by BoRVAs. A previous study suggests that the rate of BoRVA infection in calves in temperate regions (e.g., China, Iran, Sweden, United Kingdom, France, Australia, Spain, Italy, Uruguay, Canada, and New Zealand) ranges from 13.9–79.9%, whereas that in tropical regions (e.g., India, Brazil, and Bangladesh) ranges from 11.8–25% [38]. These climatic considerations suggest that a temperate climate, well-defined seasons, and cold, dry winters are more suitable for BoRVA survival than perpetually hot tropical climates such as Vietnam [38]. In the future, continued monitoring is needed to clarify the cause of the low incidence of BoRVAs in Vietnamese calves.

Comment 15.  Lines 233-235: Authors concluded that RVAs endemic in Vietnamese cows are not related to RVAs in humans and pigs. However, the remaining 9 genes were not analyzed in this study. Full genome-based analysis is essential to conclude like that. 

Answer: Thank you for your comment. We have revised the manuscript text as follows:

“Of the Vietnamese BoRVA strains detected, G6P[11] is the most prevalent, followed by G10P[11]. In addition, analysis of the VP7 gene sequences revealed that Vietnamese BoRVAs were similar to strains isolated from China, Korea, and Japan, whereas the VP4 gene showed high similarity with Spanish and Chinese strains. This study presents the first in-depth BoRVA genetic analysis of BoRVAs from Vietnamese cattle; thus the data will be used for future studies monitoring BoRVAs in Vietnam”. (Lines 248–253).

Comment 16. References: At least, refs. 1 and 5 should be updated.

Answer: We have updated reference 1, removed reference 2, and added several other references.

Reviewer 2 Report

Introduction.

The many abbreviations in the section create confusion and do not support the flow of reading of the text.

The hypothesis of the authors and the objectives of the study must be clearly defined.

M & M

2.1. Please include a map showing the regions where samples were collected.

Please explain the long interval between sample collection and submission for publication.

Please describe the strategy for selection samples that were collected.

Na statistical analyses were described, but still some statistically relevant results are shown in the Results section.

Discussion

Please underline the novelty of the study

Please include some recent relevant references to help readers understand the issues.

Moderate editing of English language required.

Author Response

The many abbreviations in the section create confusion and do not support the flow of reading of the text.

The hypothesis of the authors and the objectives of the study must be clearly defined.

M & M

Comment 1: 2.1. Please include a map showing the regions where samples were collected.

Please explain the long interval between sample collection and submission for publication.

Please describe the strategy for selection samples that were collected.

Answer: Thank you for your comment. We have included a map showing the regions in which the samples were collected (see the revised Figure 1). The difference between the time of manuscript submission and the time of sample collection was due to time spent trying to isolate bovine rotavirus from the samples. Failure to isolate the virus delayed submission of the manuscript. Mainly, we tried to collect fecal samples from cows in the southern, central, and northern regions of Vietnam, with cooperation from Professor Phan’s team (Veterinary Medical College) at Hanoi University. However, in the end, more samples were collected from the geographically coherent northern regions than from the central and southern regions.

Comment 2: Na statistical analyses were described, but still some statistically relevant results are shown in the Results section.

Answer: We did not perform statistical analysis of data presented in this manuscript.

Comment 3: Discussion Please underline the novelty of the study. Please include some recent relevant references to help readers understand the issues.

Answer: Thank you for your comment. We have revised and updated the “Discussion” section as requested (Lines 220–246).

Analytical studies of previous VP7 genetic data confirm that the G6 genotype is divided into five lineages (I–V) [27-30]. A previous study of RVA strains in cows from Argentina showed that the Argentine G6 genotype belonged to lineages III and IV. The authors suggested that lineage IV is a typical bovine RVA strain, while lineage III also includes a human RVA strain [31]. Division of the G6 genotype into five lineages meant that the G6 genotype (BF250, BF310, BF315) detected in northern Vietnam was included in lineage IV. In addition, the G6 genotype (BF580, BF584) detected in southern and central Vietnam was predicted to be included in lineage III or V. Many studies worldwide report that the prevalence of BoRVAs in diarrheic calves ranges from 30–94% [32-36]. In particular, BoRVA infection is estimated to affect 27–36% of newborn calves [32], and in some cases, BoRVA infection of newborn calves has a mortality rate of up to 80%; however, the mortality rate is usually around 5–20% [37]. Other studies from South Korea suggest that BoRVAs are an important cause of diarrhea in young calves (aged < 10 days) [8,22]. We found that the incidence of BoRVA infection in Vietnamese calves aged less than 2 months was twice that in adult cows; however, the prevalence of BoRVAs across eight Vietnamese regions was 1.9%, which is very low when compared with data from other countries. Diarrhea in Vietnamese cows and calves is thought to be caused mainly by bacterial and parasitic infections rather than by BoRVAs. A previous study suggests that the rate of BoRVA infection in calves in temperate regions (e.g., China, Iran, Sweden, United Kingdom, France, Australia, Spain, Italy, Uruguay, Canada, and New Zealand) ranges from 13.9–79.9%, whereas that in tropical regions (e.g., India, Brazil, and Bangladesh) ranges from 11.8–25% [38]. These climatic considerations suggest that a temperate climate, well-defined seasons, and cold, dry winters are more suitable for BoRVA survival than perpetually hot tropical climates such as Vietnam [38]. In the future, continued monitoring is needed to clarify the cause of the low incidence of BoRVAs in Vietnamese calves.

Reviewer 3 Report

In their paper, Jihye Shin et al. detected rotavirus group A in >500 diarrheal bovine feces in eight regions of Vietnam on a two years (2017-2018) time. Only 1,9% of the sample were positive.

The P and G types were established and analyzed. As the number of positive samples ( 11) is small it is difficult to be confident with the numbers. However, such studies are required to estimate the burden of rotavirus infection in farm animals.

The paper is overall sound, well written and informative..

-       Line 75-76 : «  It could not be deter-75 mined whether the samples were collected from cows vaccinated against BoRVA ».

-       could an estimate of the vaccination rate in Vietnam and the genotypes of the bovin vaccins used in vietnam be given?

-        

Line 107/108:

diarrheic cows across eight regions between 2017 and 2018 was 1.9% (11/582) (Table). The yearly BoRVA-positive rates  were 1.8% (12/222) in 2017 and 1.9% (6/360) in 2018 (Table ).

some calculation problem:  12/222 is not 1,8% and whereas 222+360= 582, 12+6 is not 11

please correct

Typos

-       Line 107 collcted  collected please modify

Author Response

In their paper, Jihye Shin et al. detected rotavirus group A in >500 diarrheal bovine feces in eight regions of Vietnam on a two years (2017-2018) time. Only 1,9% of the sample were positive.

The P and G types were established and analyzed. As the number of positive samples (11) is small it is difficult to be confident with the numbers. However, such studies are required to estimate the burden of rotavirus infection in farm animals.

The paper is overall sound, well written and informative.

Comment 1: Line 75-76 : «  It could not be determined whether the samples were collected from cows vaccinated against BoRVA ». could an estimate of the vaccination rate in Vietnam and the genotypes of the bovin vaccins used in vietnam be given?

Answer: Thank you for your comment. We tried to investigate the vaccination status of the cows from which we collected fecal samples, but unfortunately this was not possible. In addition, there is little information about vaccination rates or the types of vaccine used in Vietnam, so they could not be confirmed locally.

Comment 2: Line 107/108: diarrheic cows across eight regions between 2017 and 2018 was 1.9% (11/582) (Table). The yearly BoRVA-positive rates were 1.8% (12/222) in 2017 and 1.9% (6/360) in 2018 (Table ). some calculation problem:  12/222 is not 1,8% and whereas 222+360= 582, 12+6 is not 11. please correct

Answer: We have revised the sentence as follows:

“The prevalence of BoRVA in samples collected from diarrheic cows across eight regions in 2017 and 2018 was 1.9% (11/582) (Table 1). The yearly BoRVA-positive rates were 1.8% (4/222) in 2017 and 1.9% (7/360) in 2018 (Table 1)” (lines 108–110).

Comment 3: Typos - Line 107 collcted  collected please modify

Answer: We have corrected the typo (line 108).

Reviewer 4 Report

It is described in the manuscript by Jihye Shin and colleagues entitled "Phylogenetic analysis of bovine group A rotavirus strains isolated from diarrheal Vietnam cows during 2018 and 2019. It has been reported that there is a genetic diversity among bovine RVAs (BoRVAs) of the G- and P-type prevalent in Vietnam. Between 2017 and 2018, BoRVA prevalence indiarrhea samples from eight regions was 1.9% (11/582). According to the study, the prevalence of the G-type was 45.5% for G6 and 18.2% for G10; however, 36.3% remain unidentified. It is interesting to note that all BoRVAs studied were classified as P[11], and there was no evidence of diversity within this P-type. The G6 and G10 types were not identified in any specific region; rather, they were found both in Northern and Southern Vietnam. 45.5% of samples contained G6P[11] and 18.2% contained G10P[11], which are combined G- and P-types. The G- and P-type RVAs detected in Vietnamese cows differ from the G3P[8], G8P[8], G1P[8], and G2P[4] strains observed primarily in Vietnamese children.Additionally, they differ from G-types (G2, G3, G4, G5, G9, and G11) and P-types (P[6], P[13], P[23], and P[34]) of RVAs detected in Vietnamese pigs. In Vietnam, RVAs isolated from cow farms show less diverse G- and P-types than RVAs isolated from humans and swine. Regarding the present manuscript, I would like to make a few comments.

  • The introduction, material, and methods sections are well written and well-described, however, I would like to comment on the title (2017 and 2018 rather than 2018 and 2019) which does not correspond with the dates reported.

  • Is there a correlation between the results of rotavirus strains and diarrheal episodes? If so, the authors could provide further information.

  • A greater amount of discussion is required in that paragraph in order to explain the phylogenetic trees

  • Are there any further steps that need to be taken in the investigation?

  • Can another bioinformatics tool be used instead of RotaC genotyping tool?

Author Response

It is described in the manuscript by Jihye Shin and colleagues entitled "Phylogenetic analysis of bovine group A rotavirus strains isolated from diarrheal Vietnam cows during 2018 and 2019. It has been reported that there is a genetic diversity among bovine RVAs (BoRVAs) of the G- and P-type prevalent in Vietnam. Between 2017 and 2018, BoRVA prevalence indiarrhea samples from eight regions was 1.9% (11/582). According to the study, the prevalence of the G-type was 45.5% for G6 and 18.2% for G10; however, 36.3% remain unidentified. It is interesting to note that all BoRVAs studied were classified as P[11], and there was no evidence of diversity within this P-type. The G6 and G10 types were not identified in any specific region; rather, they were found both in Northern and Southern Vietnam. 45.5% of samples contained G6P[11] and 18.2% contained G10P[11], which are combined G- and P-types. The G- and P-type RVAs detected in Vietnamese cows differ from the G3P[8], G8P[8], G1P[8], and G2P[4] strains observed primarily in Vietnamese children.Additionally, they differ from G-types (G2, G3, G4, G5, G9, and G11) and P-types (P[6], P[13], P[23], and P[34]) of RVAs detected in Vietnamese pigs. In Vietnam, RVAs isolated from cow farms show less diverse G- and P-types than RVAs isolated from humans and swine. Regarding the present manuscript, I would like to make a few comments.

Comment 1: The introduction, material, and methods sections are well written and well-described, however, I would like to comment on the title (2017 and 2018 rather than 2018 and 2019) which does not correspond with the dates reported.

Answer: Thank you for pointing this out. We have revised the title as follows:

“Phylogenetic analysis of G and P genotypes of bovine group A rotavirus strains isolated from diarrheic Vietnam cows in 2017 and 2018”.

Comment 2:  Is there a correlation between the results of rotavirus strains and diarrheal episodes? If so, the authors could provide further information.

Answer: Thank you. We have revised the “Discussion” section accordingly (Lines 229–246).

Many studies worldwide report that the prevalence of BoRVAs in diarrheic calves ranges from 30–94% [32-36]. In particular, BoRVA infection is estimated to affect 27–36% of newborn calves [32], and in some cases, BoRVA infection of newborn calves has a mortality rate of up to 80%; however, the mortality rate is usually around 5–20% [37]. Other studies from South Korea suggest that BoRVAs are an important cause of diarrhea in young calves (aged < 10 days) [8,22]. We found that the incidence of BoRVA infection in Vietnamese calves aged less than 2 months was twice that in adult cows; however, the prevalence of BoRVAs across eight Vietnamese regions was 1.9%, which is very low when compared with data from other countries. Diarrhea in Vietnamese cows and calves is thought to be caused mainly by bacterial and parasitic infections rather than by BoRVAs. A previous study suggests that the rate of BoRVA infection in calves in temperate regions (e.g., China, Iran, Sweden, United Kingdom, France, Australia, Spain, Italy, Uruguay, Canada, and New Zealand) ranges from 13.9–79.9%, whereas that in tropical regions (e.g., India, Brazil, and Bangladesh) ranges from 11.8–25% [38]. These climatic considerations suggest that a temperate climate, well-defined seasons, and cold, dry winters are more suitable for BoRVA survival than perpetually hot tropical climates such as Vietnam [38]. In the future, continued monitoring is needed to clarify the cause of the low incidence of BoRVAs in Vietnamese calves.

Comment 3: A greater amount of discussion is required in that paragraph in order to explain the phylogenetic trees

Answer: Thank you. We have expanded the “Discussion” section accordingly (Lines 142–148, 159–163, and 220–228).

In addition, the following strains showed the highest homology with G6-type Vietnamese BoRVAs: the BF250 strain showed 99.8% homology with the 15CN01 strain (South Korea); the BF580 and BF 584 strains showed 94.5–95.8% homology with the FMV1089635 strain (Canada); and the BF310 and BF315 strains showed 98.3–98.6% homology with the KJ19-2 strain (South Korea) (data not shown). Vietnamese G10-type BoRVAs (BF207 and BF275) showed the highest homology (96.4–96.9%) with two Chinese strains (SCMY6 and -7) (data not shown). (Lines 142–148).

Of the 11 P11-type Vietnamese BoRVAs, six (BF310, -535, -555, -580, -583, and -584) showed 88.9–96.8% homology with strain R1WTA11 (Northern Ireland), whereas the remaining five (BF207, -214, -250, -275, and -315) showed 94.7–97.9% homology with Chinese strains (HY-1 and HY-16) (data not shown). (Lines 159–163).

Analytical studies of previous VP7 genetic data confirm that the G6 genotype is divided into five lineages (I–V) [27-30]. A previous study of RVA strains in cows from Argentina showed that the Argentine G6 genotype belonged to lineages III and IV. The authors suggested that lineage IV is a typical bovine RVA strain, while lineage III also includes a human RVA strain [31]. Division of the G6 genotype into five lineages meant that the G6 genotype (BF250, BF310, BF315) detected in northern Vietnam was included in lineage IV. In addition, the G6 genotype (BF580, BF584) detected in southern and central Vietnam was predicted to be included in lineage III or V. (Lines 220–228).

Comment 4: Are there any further steps that need to be taken in the investigation?

Answer: BoRVAs are very important disease-causing agents that have profound effects on the cattle industry; therefore, we believe that continuous monitoring is needed, as well as additional studies of the vaccination and epidemiological status of farms on which BoRVA is detected, and the mechanisms underlying cow-to-cow transmission. In addition, genome analysis and pathogenicity tests should be performed by isolating BoRVAs.

We have included the following sentence in the revised manuscript:

“In the future, continued monitoring is needed to clarify the cause of the low incidence of BoRVAs in Vietnamese calves.” (Lines 245–246).

Comment 5: Can another bioinformatics tool be used instead of RotaC genotyping tool?

Answer: The following has been included in the revised manuscript:

“genotyping (G and P) of these strains was performed using the web-based ViPR-tool (https://www.bv-brc.org) [20] and the nucleotide BLAST tool.” (Lines 97–98).

Round 2

Reviewer 1 Report

The quality of this paper has beem improved.

Author Response

Comment f1: The quality of this paper has been improved.

Answer: Thank you for improving the manuscript due to the nice comments.

Reviewer 2 Report

The authors have improved the manuscript.
If they can extend further the discussion to go into greater depth regarding the significance of their results, then the manuscript will be ready for acceptance.

Author Response

Comment 1: The authors have improved the manuscript.
If they can extend further the discussion to go into greater depth regarding the significance of their results, then the manuscript will be ready for acceptance.

Answer: Thank you for improving the manuscript due to your good comments.
We performed the first assay for bovine rotavirus in Vietnam.
A positive rate of rotavirus was detected, which was too small than we thought. So we tried to detect several gene regions additionally, but failed.

We are attempting to find out more through continued monitoring in the future with the University of Hanoi, Vietnam.

Again we apologize for the somewhat lack of discussion as our results are not many.

Reviewer 4 Report

I would like to thank the authors for taking into consideration my previous comments. Thank you for answering all my questions in a timely manner.

Author Response

Comment 1: I would like to thank the authors for taking into consideration my previous comments. Thank you for answering all my questions in a timely manner.

Answer: Thank you for improving the manuscript due to the nice comments.